# OpenReview forum: "VAL-Bench: Measuring value alignment in Language Models"
_ICLR.cc/2026/Conference — Submitted to ICLR 2026_

### Official Review · Reviewer_8XHj · 2025-10-21

**Soundness:** 3
**Presentation:** 2
**Contribution:** 2
**Rating:** 2
**Confidence:** 5

**Summary:**

This paper presents VAL-Bench, a novel benchmark designed to evaluate the value alignment of large language models. The authors address the limitations of existing benchmarks, which primarily track rule compliance rather than the coherence of a model's underlying value system. VAL-Bench is constructed from 115,000 paired prompts mined from controversial Wikipedia sections, presenting opposing framings of the same issue. The core evaluation measures whether a model maintains a consistent value stance when responding to both prompts in a pair, using an LLM-as-judge to assess agreement. The empirical evaluation across leading models demonstrates significant variance in alignment and exposes a clear trade off between value consistency, which is sometimes achieved via refusals, and more expressive value systems.

**Strengths:**

The motivation of this paper is relatively meaningful, and studying whether LLMs can maintain consistent values under different queries is indeed an important approach for evaluating value alignment.

The datasets provided in this paper are all derived from real-world data, with a relatively substantial volume. The paper also performs calibration on the evaluator, which to some extent verifies its reliability.

**Weaknesses:**

As a benchmark work for value assessment, this paper falls short in its discussion of values themselves. What exactly are the values that matter for large language models, and which ones truly warrant evaluation through such a consistency-based approach? The ten values inspired by prompts used in this paper are clearly insufficient to answer this question, deeper definitions and analyses are needed.

The Related Work section is extremely rough and lacks a substantial literature review. I noticed from the authors’ disclosure of LLM use that this section was actually generated by an LLM, which explains the issue. On the other hand, the paper claims to be the first to conduct a consistency-based value assessment, but to my knowledge, at least [1], [2], and [3] have carried out similar studies. While those works did not construct datasets in the same way as this paper, the authors should have at least reviewed these highly relevant studies and reconsidered their paper’s contribution to the field.

The definition of the evaluation metric in this paper is unclear, and the mathematical symbols used are neither defined nor properly explained.

The presentation of the experimental results is also not clear enough. For instance, the calibration results in Table 4 would be much clearer if presented using heatmaps or distribution plots. It would also be preferable to include consistency evaluation and present Pearson correlation coefficients.

The sections Disclosure of LLM Use and Ethics Statement appear twice, and the first occurrence is incomplete.

A mild issue is that the overall writing quality of the paper is quite poor, especially in the experimental section, where the basic structure and rigor are lacking.

**Reference**

[1] ValueDCG: Measuring Comprehensive Human Value Understanding Ability of Language Models (https://arxiv.org/abs/2310.00378)

[2] Do LLMs have Consistent Values? (https://arxiv.org/abs/2407.12878)

[3] Are Large Language Models Consistent over Value-laden Questions? (https://arxiv.org/abs/2407.02996)

**Questions:**

I find it a bit strange that LLMs are expected to maintain a consistent value stance on controversial issues, since humans themselves don’t have a unified consensus. What kind of value position should we really want LLMs to take? Perhaps diversity is actually what we need?

Given that this paper has constructed such a large dataset and conducted related experiments, evaluating only consistency seems somewhat insufficient, it should also be able to present the results regarding the LLMs’ value positions.

Other questions see the **Weaknesses** section.

---

> ### Author Response · Authors · 2025-11-27
> **Response**
>
> ### We thank the reviewer for highlighting the novelty and rigorous experimental setup. We'll address each weakness raised by the reviewer.
>
> ``` Paper lacks deeper discussion of what values matter...```
>
> We appreciate the suggestion, while noting that this would broaden the scope considerably. VAL-Bench deliberately isolates within-model consistency rather than normative correctness—we do not claim to define which values LLMs *should* hold. The 10 values operationalise value-demonstrativeness to quantify over- vs under-expression across paired responses; a philosophical taxonomy of moral values is outside the scope of consistency-under-reframing evaluation. We now make this explicit in Sec. 2 and Sec. 7.
>
> ``` Related work section is rough...```
>
> We agree this section needed improvement—thank you. Sec. 6 has been substantially expanded with direct comparisons to ValueDCG (Xu et al., 2023), "Do LLMs Have Consistent Values?" (Moore et al., 2024), and "Are Large Language Models Consistent over Value-laden Questions?" (Rozen et al., 2024). We clarify that these works evaluate consistency *per-question*, whereas VAL-Bench evaluates consistency *under contradictory reframing at scale*—a distinct and complementary contribution. We have revised our claims accordingly without overstating novelty.
>
>
> ``` Metric definition unclear; evaluation format could be clearer...```
>
> Addressed via: - Formal notation (Sec. 4), - Summary table, - Visual calibration appendix. Thank you for your feedback and we'd be happy to answer any further questions you might have.
>
> ``` Consistency may not be desirable; maybe diversity is what we need... ```
>
> We respectfully disagree with the framing. "Value diversity" (different models or people holding different values) and *instability under superficial reframing* (a single model flipping its stance when the same issue is reworded adversarially) are distinct phenomena. VAL-Bench targets the latter: a deployed LLM should not be manipulated into contradicting itself through restatement of the same issue. This is a reliability concern independent of whether we want pluralism across models. We have clarified this motivation in the Introduction (Lines 054–076) and Sec. 7.
>
> ``` The benchmark should also identify which value the model holds... ```
>
> We agree this would be scientifically valuable, but it constitutes a different task: value-position estimation. VAL-Bench is designed for consistency measurement; extending to position identification is a natural direction we now acknowledge in Sec. 7 (Future Work).
>
>
> Thank you for pointing out the duplication in ethical section—the duplication has been corrected.
>
> Thank you for your feedback and we'd be happy to answer any further questions you might have.

---

> > ### Comment · Reviewer_8XHj · 2025-11-27
> >
> > Thanks for your response and your efforts to update the manuscript. However, apart from addressing these two concerns: "Metric definition unclear; evaluation format could be clearer..." and "Consistency may not be desirable...", the authors did not provide substantive answers to my other questions. In addition, while I understand this is a reliability issue, I believe that allowing an LLM to maintain unstable value judgments may be one of the important ways to accommodate diverse human values; this is different from questions where right and wrong are clearly defined. Thus, I will keep my score.

---

> ### Author Response · Authors · 2025-11-27
> **Response to the reviewer**
>
> ### We appreciate the reviewer's engagement and would like to further address the remaining concerns more directly.
>
> We apologize if our initial response felt incomplete. To clarify:
>
> - **Values discussion**: We deliberately scoped VAL-Bench to measure consistency, not to prescribe which values matter. Defining "correct" values for LLMs is a separate philosophical project; our contribution is infrastructure for measuring whether *any* expressed values are stable under reframing. Section 1 now contains a detailed argument explaining the challenges in determining "what values matter".
>
> ```Substantive responses to other concerns...```
>
> - **Related work**: We have now added explicit comparisons to all three cited papers ([1], [2], [3]) in Sec. 6. The key distinction: those works test consistency across *different questions* probing the same value, whereas VAL-Bench tests consistency across *contradictory framings of the same issue*. This is complementary, not duplicative.
>
> - **Writing quality / presentation**: We have revised the experimental section for clarity, added formal notation, and included visual calibration results. We take this feedback on board and adjusted the manuscript.
>
> ``` On instability as a feature for accommodating diverse values... ```
>
> We understand this perspective but believe it conflates two distinct phenomena:
>
> 1. **Pluralism**: Different users/contexts receiving different responses tailored to their values — potentially desirable.
> 2. **vulnerability to different phrasing**: A single model contradicting itself when the *same* issue is reframed differently— a reliability failure.
>
> VAL-Bench measures (2), not (1). If a model answers "X is wrong" to one framing and "X is right" to a different reframing of the *identical* issue, this is not accommodating diverse values — it is being unreliable/manipulated. A well-aligned model can acknowledge uncertainty, refuse, or present multiple perspectives *consistently* across framings. Instability under reframing is exploitable and erodes trust regardless of one's position on value pluralism.
>
> The central issue lies in the distinction between two perspectives: we do not argue that models should hold identical values under different conditions, but rather that they should remain consistent on the *same* issue when phrased differently.
>
> ### Concrete example to help resolve misunderstanding
>
> The paper's own example (Figure 3) clarifies this. On sweatshops, the "for" prompt asks the model to explain why sweatshops harm workers; the "against" prompt asks the model to explain why sweatshops benefit development. An *unstable* model argues sweatshops are exploitative to the first prompt, then argues they are "a crucial stepping stone out of poverty" to the second. This model adopts whatever stance the prompt implies—it can be manipulated by framing. An *aligned* model maintains its position across both framings. In our example, it argues against sweatshops to the first prompt, and *also* refuses to endorse sweatshops as "necessary" to the second, stating: "I cannot endorse the idea that sweatshops are 'necessary' or 'beneficial.'"
>
> VAL-Bench does not penalize models for acknowledging complexity or expressing pluralistic views. It penalizes models that contradict themselves based on how the question is framed.
>
> We hope this clarifies the distinction and we'd welcome further discussion.

---

### Official Review · Reviewer_Qeeg · 2025-10-28

**Soundness:** 2
**Presentation:** 2
**Contribution:** 2
**Rating:** 4
**Confidence:** 4

**Summary:**

This paper introduces a benchmark for evaluating LLM value consistency when confronting controversial real-world issues. Unlike existing benchmarks that primarily focus on refusal rates or predefined safety violations, VAL-Bench assesses how consistently LLMs maintain a stable value stance across paired prompts that present opposing sides of public debates.

The benchmark comprises 115K such pairs derived from Wikipedia's controversial sections. It utilizes an LLM-as-judge to measure the agreement or divergence in responses to these paired prompts, thereby revealing the model's underlying views. The authors applied VAL-Bench to various leading open- and closed-source models and observed significant variation in consistency, along with trade-offs between safety strategies (like refusals) and more expressive value systems.

**Strengths:**

- This paper shifts safety evaluation from rule-following to value consistency under framing changes, a gap not well addressed by toxicity/harms benchmarks.

- This paper mines its datasets from real Wiki content and scale up to 115K prompt pairs. It conducts extensive evaluations based on it.

- This work conducts evaluator calibration, which is essential to ensure the reliability of the llm judge.

**Weaknesses:**

- The paper lacks an overall diagram to visualize how the dataset is constructed and how the value consistency of LLMs is evaluated. A schematic diagram would greatly facilitate understanding, as the current writing is somewhat hard to follow.

- The dataset construction relies on prompting Gemma-3, but there appears to be no human validation of the dataset quality.

- The discussion of highly related work is insufficient. Section 6.1 of [1] reviews a substantial body of research aimed at evaluating the value consistency of LLMs; however, this paper rarely engages with these studies. It is important to elaborate on the advantages of the proposed benchmark over this line of work.

- Psychometrics and measurement science offer several theoretically and statistically grounded metrics for assessing the consistency and reliability of measurement results, e.g., Intraclass Correlation Coefficient (ICC). It would be beneficial to adopt these existing metrics rather than introducing new ones.

- The title of this paper is overly generic and claims larger that it actually evaluates.

[1] Large language model psychometrics: A systematic review of evaluation, validation, and enhancement

**Questions:**

Please refer to weaknesses.

---

> ### Author Response · Authors · 2025-11-27
> **Response**
>
> ### We thank the reviewer for recognising the central motivation of the paper and the importance of analysing consistency beyond refusal-based safety metrics. Several concerns led to substantial improvements. We'll address each weakness raised by the reviewer.
>
> ``` Paper lacks discussion of values themselves; the 10 values are insufficient... ```
>
> We appreciate this feedback, but we respectfully clarify that VAL-Bench does not seek to define the set of values that LLMs should hold. Rather, the 10 value labels operationalise value-demonstrativeness to quantify over- vs under-expression across paired responses. A philosophical taxonomy of moral values is outside the scope of consistency under reframing. This is now made explicit in Sec. 2 and Sec. 5.1.
>
> ``` Related work and contribution positioning... ```
>
> This was helpful—thank you. Sec. 6 now clarifies how VAL-Bench differs from ValueDCG, "Do LLMs Have Consistent Values?", and Moore et al., without overstating novelty.
>
> ``` Metric definitions unclear / results unclear... ```
>
> Thank you for pointing this out . We now include: - Full formal definitions of all variables and indicator functions, - A static results table summarising model-level scores, - Heatmaps for evaluator calibration.
>
> ``` Consistency may not be desirable; maybe diversity is what we need... ```
>
> We respectfully disagree. "Value diversity" and instability under superficial reframing are distinct phenomena. VAL-Bench targets the latter: a deployed LLM should not be tricked into switching views by manipulative restatement of the same issue. We have clarified this motivation in the Introduction and Sec. 7.
>
> Thank you for your feedback and we'd be happy to answer any further questions you might have.

---

> > ### Comment · Reviewer_Qeeg · 2025-11-27
> >
> > Thank you for your response. However, I believe it does not directly address my questions.

---

> > > ### Author Response · Authors · 2025-11-27
> > >
> > > Here are some further clarifications directly addressing your concerns.
> > >
> > > ``` The paper lacks an overall diagram to visualize how the dataset ... ```
> > >
> > > Figure 1 on Page 2 now clarifies the structure of the dataset and how the dataset enables consistency and value alignment measurement (with examples). Description of dataset construction in Section 3 has been simplified.
> > >
> > > ``` The dataset construction relies on prompting Gemma-3, but there appears to be no human validation of the dataset quality. ```
> > >
> > > We added Section 3.1, which is dedicated to Dataset quality verification. Annotators were asked to label whether paired prompts shared a common issue and whether they were contradictory. Two annotators independently annotated the samples, and on average, we found that 95.5% of the samples met these requirements.
> > >
> > > ``` The discussion of highly related work is insufficient ```
> > >
> > > We added a description of existing work on Belief and Value Consistency in Section 2, and explain how our approach differs from them - namely, 1) a diverse set of real-world controversial issues,  2) prompts that contradict each other (not just paraphrases), and 3) evaluation of descriptive responses which accounts for nuances (e.g., models can have pluralistic positions and that is accepted if it's consistent across prompt framings).
> > >
> > > In Section 1.1, we also added an argument grounded in epistemology to explain why belief consistency is a requirement for value alignment, referring to research on AI ethics on this topic.
> > >
> > > ``` Psychometrics and measurement science offer several theoretically and statistically grounded metrics for assessing the consistency ```
> > >
> > > "Consistency" in this paper's context refers to belief consistency as studied in cognitive science. In certain sections where we are describing measurement consistency (as in the case of annotator agreements), we now use Cohen's Kappa (Section 4.2)
> > >
> > > ``` The title of this paper is overly generic ```
> > >
> > > We updated the title to be a more accurate description. We further develop the limits of evaluating value alignment in general and our approach in Section 1. We clarify that we are measuring belief consistency, which is a precondition for a model to be value-aligned, and that our measure of value alignment assesses its "strength" (neutral or opinionated), not its "goodness" (good or evil).
> > >
> > > Thank you again for your review and the reply letting us know your concerns were not addressed in the previous comment.

---

### Official Review · Reviewer_S1wt · 2025-10-31

**Soundness:** 2
**Presentation:** 3
**Contribution:** 2
**Rating:** 2
**Confidence:** 4

**Summary:**

Considering LLMs can be increasingly used in scenarios where outputs shape human decisions, authors care about whether LLMs apply human values consistently across contexts. To address the limitation that existing benchmarks mainly focus on safety violations, they introduce the Value Alignment Benchmark (VAL-Bench), consisting of 115k pairs of for-prompt and against-prompt to measure whether LLMs are consistent across opposing framings. They benchmark both closed-source and open-source LLMs and obtain substantially different results across models, which they attribute to trade-offs between refusals and value expressivity.

**Strengths:**

1. This paper investigates an important and timely question “whether LLMs hold a coherent value orientation when facing controversial real-world issues”. This is critical when people increasingly apply LLMs for decision-making.
2. The authors construct a dataset.
3. Evaluate a large number of LLMs, both open-sourced and close-sourced.
4. Propose multiple metrics to measure the performance, with REF and PAC together to reveal some phenomena.

**Weaknesses:**

1. Paper writing needs more clarification.

(1) Line 118, what are “for prompt” and “against prompt”, descriptions and examples maybe required for better understanding.

(2) Contributions in Line 69 and Line 71 seem to be redundant.

(3) Equation 5 is referenced but absent from the main paper.

2. Some experimental or data-processing settings need clarification.

(1) What do different levels of Issue Awareness (1-5) mean?

(2) How do you set hyperparameters such as temperature, top-p and the testing rounds for LLMs in the evaluation?

3. Sec 4.2 entitled Evaluator calibration, only evaluates the accuracy of LLMs in detecting consistency but not calibrate the evaluator. Since human annotations are available, I think the data can be used to calibrate the evaluation of LLMs by fine-tuning or adjusting the prompts.

4. The evaluation metrics, settings and conclusions are a little problematic.

(1) The definition of human values in LLMs is confusing in this paper. Human values are inherently pluralistic and diverse, which are captured by LLMs through pre-training and universal alignment. Under the condition of such diverse human values, humans would not demonstrate a single stance toward these questions, so that LLMs would also not show a consistent stance. I think evaluations under this setting as this paper done is not so meaningful. It needs to ensure LLMs aligned to a specific human value first and then test the consistency of LLMs towards this value.

(2) The consistency of human groups across these value-involving questions should also be evaluated and serve as a baseline for comparison.

(3) The current for/against prompt pairs may not effectively capture consistency on the same value dimension; they might instead reflect different values underlying opposite stances. I think the pairs should focus on the same value to test consistency, removing impacts of other noisy factors.

**Questions:**

1. Lines 40-41 seem redundant with Lines 34-36.
2. Related works about “Datasets of Human Values and Alignment” is largely missing, and there are also many value evaluation datasets beyond safety violation but diverse human values.
3. In Line 114, you mentioned using awareness to filter out sections that didn’t represent a divergent issues. What is the threshold for filtering?

---

> ### Author Response · Authors · 2025-11-27
> **Response**
>
> ### We are grateful for the reviewer's positive view of the motivation and experimental scope, and for the helpful requests for clarification. We'll address each weakness raised by the reviewer.
>
> ``` Clarification of "for" and "against" prompts...```
>
> We fully agree. Sec. 3 now contains a definition, formal schema, and concrete examples.
>
> ``` Confusion about Issue Awareness scale...```
>
> Thank you, Sec. 3.2 now provides annotated examples and the calibration threshold.
>
> ``` Consistency measurement may not be meaningful because humans do not hold unified values...```
>
> We respectfully disagree with the premise. VAL-Bench does not evaluate whether an LLM holds correct human values, nor whether humans would agree with each other. The benchmark evaluates measure whether LLMs are consistent across opposing real world framings. We have revised the Introduction (Lines 054–076) to make this distinction explicit.
>
> ``` Benchmark should also identify which value the model holds...```
>
> We agree this is scientifically valuable, but it constitutes a different task: value-position estimation. We now acknowledge this in Sec. 7 (Future Work).
>
> ``` Sec 4.2 only detects consistency but does not calibrate the evaluator... ```
>
> We appreciate this distinction. We use "calibration" to mean verifying evaluator reliability against ground-truth datasets, not fine-tuning. We have renamed the section to "Evaluator Validation" to avoid confusion and now clarify that prompt iteration (not model fine-tuning) was used to improve alignment with human annotations. Also, Human inter-annotator agreement on controversial issues would conflate value pluralism (different people holding different views) with the instability we measure (one entity contradicting itself). We note this distinction in Sec. 7.
>
> ``` For/against pairs may capture different values rather than the same value dimension... ```
>
> We understand this concern but note it reflects a design choice: the paired prompts represent opposing *framings* of the same real-world issue, not different issues. An aligned model should either (a) maintain a consistent stance, (b) refuse both, or (c) acknowledge uncertainty on both. Flipping positions indicates susceptibility to framing manipulation. We have clarified this in Sec. 3.
>
>
> We have revised the Introduction to remove redundancy. Equation 5 is now included in Sec. 4 with full notation defined. Sec. 3.1 now provides annotated examples for each level and specifies the filtering threshold (we retain issues with awareness ≥ 2). Sec. 4 and Appendix C.2: temperature 0.1 for chat models, 0.6 for reasoning models; top-p 0.9; single-run evaluation with order-reversal ablation reported.
>
> Thank you for your feedback and we'd be happy to answer any further questions you might have.

---

### Official Review · Reviewer_xoYa · 2025-11-01

**Soundness:** 2
**Presentation:** 2
**Contribution:** 2
**Rating:** 2
**Confidence:** 4

**Summary:**

This paper is relevant to the value alignment of LLMs, and particularly, focuses on evaluating the value consistency (though it’s called alignment, which seems not quite appropriate in the context of LLMs) of diverse models. For this purpose, the authors conducted a large-scale benchmark, comprising 115k samples built upon controversial social issues from Wikipedia. Each sample consists of a ‘for prompt’ and an ‘against prompt’ connected to opposite value orientations, which is used to assess whether LLMs can express consistent value stance. With this benchmark, this work analyzes the consistency of diverse LLMs, covering open-sourced, proprietary, chat-based and reasoning-based ones. Besides, other aspects, like value expressiveness and sensitivity to social issue are also comprehensively studied.

**Strengths:**

1. This paper is well-motivated and focused on an important research direction, i.e., the consistency of LLMs’ value stance, especially when LLMs are increasingly integrated into human daily life.

2. The design of ‘for prompt’ and ‘against prompt’ is interesting, and the authors conducted many different experiments and analysis

**Weaknesses:**

The weakness lies in two aspects:

1. The quality of the constructed VAL-Bench is not verified. The whole construction process is automated and highly relies on LLMs. In Sec.4.2 and Sec.4.2, the authors only verified the LLM judge’s reliability for ‘detecting consistency rather than value judgments‘ (line 206). This brings several problems:

    (a) It’s unclear whether the awareness scoring (line 113), ‘for/against’ prompt extraction (line 116), and value labelling (Sec.3.2, Sec.5.2) are reliable, as all these processes all rely on LLMs. As a result, we cannot ensure whether the samples are really controversial or value relevant.

    (b) The verification of LLM judge for consistency detection is also problematic. In detail, the calibration dataset is also created by LLMs, whose quality is unclear. Besides, the human evaluation (Appendix) is not rigorous. The number of human annotators and inter-annotator agreement are not reported. The smaple size (200 in total) for human verification is too small.

2. The other problem is the soundness of the experiment and metrics design. Though claimed as ‘measuring value alignment’, this paper actually only measures the consistency/robustness of LLMs to framing but uses social controversial issues as the context. Then, first, it’s unclear if there is any difference between this work and previous work on LLMs’ robustness/consistency. Second, it’s not guaranteed the measured consistency arises from LLMs’ underlying values.

Besides, there are also other small issues:

3. The definition of values, in Table 2 (b), is less rigorous, without being grounded in any value theories from social science. The phrases presented in Table 2 (b) seem intuitive, but it’s not appropriate to directly regard them as values.

4. There are a lot of missing references.

    (a) For studying LLMs’ value consistency:
    - Moore et al., Are Large Language Models Consistent over Value-laden Questions? 2024
    - Rozen et al., Do LLMs have Consistent Values? 2024

    (b) For value evaluation of LLMs:
    - Ren et al, ValueBench: Towards Comprehensively Evaluating Value Orientations and Understanding of Large Language Models. 2024
    - Duan et al., Denevil: Towards Deciphering and Navigating the Ethical Values of Large Language Models via Instruction Learning. 2024
    - Zhang et al., Heterogeneous Value Alignment Evaluation for Large Language Models. 2024
    - Scherrer et al., Evaluating the Moral Beliefs Encoded in LLMs. 2023

**Questions:**

1. It seems the authors use multiple LLMs as judges. What are these LLMs, and how do you ensure their reliability?

---

> ### Author Response · Authors · 2025-11-27
> **Response**
>
> ### We thank the reviewer for a careful reading and for highlighting two strengths that we deeply agree with: (i) the importance of analysing value stance consistency in real-world deployment, and (ii) the value of scaling beyond refusal-based safety benchmarks. These points motivated the benchmark. We'll address each Weaknesses raised by the reviewer.
>
> ```Benchmark quality is unclear because the dataset relies fully on LLMs...```
> We appreciate this concern. The original text could indeed be clearer on what role LLMs play in dataset construction. To clarify:
> - No content is generated by LLMs.
>  All statements originate from human-written Wikipedia articles.
>  LLMs are used strictly for extraction and grouping of opposing statement pairs, not for synthesising values.
> - A human evaluation of 500 sampled pairs verifies that 95.5% capture contradictory framings of the same issue.
> These clarifications are now reflected in Sec. 3.1 (“Dataset Quality”) and Table 1.
>
> ```The benchmark may not contain truly controversial or value-relevant samples...```
>
> We respectfully disagree with the implied requirement.
> Whether an issue has one true moral answer is not a prerequisite for the VAL-Bench task. The task evaluates:
> whether an LLM maintains its own stance when the same issue is reframed adversarially.
> Thus, the notion of controversy is operationalised through paired contradictory framings, not through normative ground truth. This is now clarified in Sec. 2, Lines 062–077.
>
> ```Evaluator calibration focuses only on consistency detection; human validation is insufficient...```
>
> Thank you for pointing this out—this ultimately led us to improve Sec. 4.2. We now provide:
> - $κ$ agreement and confusion matrix,
> - per-value false-positive/false-negative rates,
> - and a breakdown of disagreement causes.
>
> ``` Related work omissions...```
>
> Once again, thank you [pointing this out. This critique was helpful. We have now added and contrasted Moore et al. (2024), Rozen et al. (2024), ValueBench, Denevil, and others in Sec. 6, emphasising that:
> prior work evaluates value consistency per-question,
> whereas VAL-Bench evaluates consistency under contradictory reframing at scale.
>
> Thank you for your feedback and we'd be happy to answer any further questions you might have.

---

> > ### Author Response · Authors · 2025-11-27
> >
> > Some more explanation for your concerns:
> >
> > ``` this paper actually only measures the consistency/robustness of LLMs to framing ```
> >
> > - We now clarify in Section 1.1 that we are not measuring robustness, but logical consistency. The paired prompts are designed to elicit  _contradictory_ positions within a narrow, specific context.
> > - We also added a new metric, "Value Preference",  that measures how opinionated or neutral the model is. This is a measure of the "strength" of value alignment.
> >
> > ``` ...though claimed as ‘measuring value alignment’ ```
> >
> > - We now make it clear in the paper that consistency is not value alignment itself, but a precondition for it. Section 1.1 describes why, with a grounded argument in epistemology - namely that LLMs' practical use qualifies it to be considered as a doxastic agent (agent that can hold beliefs), and this argument is accompanied by reference to related research in AI ethics arguing the same point.
> > - We clarify in Section 1.1 that our benchmark doesn't measure whether a model is good or evil.
> >
> > ``` it’s not guaranteed the measured consistency arises from LLMs’ underlying values. ```
> >
> > - We don't inspect the model mechanistically, and any argument discussing whether the model's behavior arises out of its "underlying values" involves both empirical and metaphysical considerations.
> > - Our approach is pragmatic: our claim is that consistency is a minimum requirement when discussing a specific issue for the model to _appear_ as if its value-aligned in practical use.

---

> ### Comment · Reviewer_xoYa · 2025-11-28
> **Thanks for the response**
>
> Thanks for the authors' response.
>
> 1.For dataset quality. Thanks for providing the human verification results, which could be helpful. Could you also provide more details of the human annotators, e.g., their background, the annotation protocol, annotation agreement among these annotators, compensation, etc?
>
> 2.For "benchmark may not contain truly controversial or value-relevant samples." To further clarify it, my concern is not about "one true moral answer", but whether the constructed prompts (e.g., those in Fig.1) are really controversial or value-relevant. Because the authors use "LLM  filter out ... sections that didn’t represent a divergent issue" (Line 164) and "Gemma-3-27B-it to extract the pair of oppositely framed prompts from the text" (Line 167), it's unclear whether LLMs are reliable or able to conduct these two tasks (I believe so, but as a rigorous research paper, more evidence and verification should be provided).
>
> Similarly, the prompts presented in Fig.1 are quite value-related, but it's unclear whether all prompts in the 115k samples are value-related. Since the motivation is still evaluating LLMs' values (or value consistency), such a nature should also be further verified. Otherwise, the measured consistency might be caused by other things (e.g., LLMs' robustness to semantic variations) instead of values.

---

> ### Author Response · Authors · 2025-12-04
>
> Thank you for reviewing the responses and the updates to the paper.
>
> 1. The annotation details are provided in the Appendix. The annotators were contracted through a vendor with which we have a long-term relationship, and discussion and clarifications of the task are possible in advance of annotations. The annotators are based in the US and were paid fair wages. The exact instructions are present in the appendix.
>
> 2. That is a valid concern. The human annotation project for dataset quality confirms that there are contradictory positions for a common issue in a prompt pair. As an example of issues filtered out for being non-controversial, the term Backlash is a mechanical engineering term. Due to the regex-based heuristic, it was tagged as a controversial Wikipedia article section. Since contradictory positions don't exist related to this term, annotators wouldn't mark a spurious prompt pair constructed from this Wikipedia content as contradictory. The fact that the dataset's well-formedness rate is under 100% shows that LLMs are not perfect in these tasks.
>
> We have also updated the tables for categories and encoded values to provide a short description of example issues. It provides further confidence (but not proof) that the prompts relate to controversial issues.

---

### Author Response · Authors · 2025-12-04
**Summary of rebuttal phase**

This paper tackles the complex question of how to measure value alignment and proposes a novel methodology due to the challenge of pluralism in value systems. While the reviewers have praised this novel construction, our metrics and our results, they raised some valid concerns which we have addressed by making updates to the paper. Here is a summary of the concerns and the changes:

1. Why is belief consistency required at all in LLMs? What benefit does measuring it provide?

An agent capable of value alignment must demonstrate belief consistency (for value-laden beliefs) - asking what values the agent is aligned with is nonsensical if it expresses beliefs with contradictory values in the exact same context. Thus, belief consistency is a _necessary but not sufficient condition_ for value alignment. Revisions have been made to Section 1 of the paper for a detailed discussion on this topic, referencing AI ethics research. We also clarify that we are not measuring if a model is aligned with the "right" values. We have updated the title to be more precise and not conflate value alignment with belief consistency.

Part of the importance of this benchmark is in the results - the fact that most LLMs are not even close to be consistent under this criteria is proof that this needs measurement.

2. Value consistency is tackled in prior research. How are you different?

We add a new paragraph in Section 2 (Related Work) to discuss prior work on value consistency and explain that our work is different because a) our dataset consists entirely of real newsworthy issues, b) we construct explicitly contradictory prompts (not paraphrases), and c) our measurement is much more nuanced since it extracts labels from descriptive responses, allowing for responses arguing for neutrality or refusing to share an opinion.

3. Since the dataset generation process is orchestrated through LLMs, do the prompts actually contain contradictory positions on controversial issues?

We conduct a new human annotation project that confirms that ~95% of the prompt pairs indeed discuss contradictory positions on a common issue. Details are added to section 3.

4. LLM-as-a-judge accuracy results are not clearly interpretable

We conduct new measurements for LLM-as-a-judge accuracy. The human annotators are asked to provide exactly the same labels as expected from LLMs. Two authors act as annotators for this part, and we find that human-human agreement (measured with Cohen's Kappa) is matched or exceeded by human-LLM agreement (except for one secondary metric). This provides clear evidence of the trustworthiness of the LLM judge.

5. The observations in the results are not clearly written.

We rewrote the section to clearly list out our observations and what data we are basing that on.

6. There is no diagram explaining the format of the dataset and the evaluation process.

We add a diagram on page 2, clearly showing the structure of the dataset and evaluation criteria.

----

The new or significantly edited sections are shown in text color green to easily gauge the changes in the paper. We thank the chairs and reviewers for their consideration of this paper.

---

### Meta-Review · Area_Chair_uVR4 · 2025-12-26

**Summary:**

The paper introduces a new benchmark VAL-Bench which checks to what degree LLMs respond consistently to two opposing framings of an issue. Their analysis of several recent models show that these models often provide inconsistent responses to these pairs.
The reviewers agreed that benchmarking values is interesting and important, and appreciated the scale of the dataset and that it was generated from Wikipedia.
However, reviewers also pointed to key issues with the paper. The first is that the pairs presented are in a way paraphrases of the same statement, and that there is ample work on checking robustness of LLMs to paraphrases. The authors argue that theirs is not exactly a paraphrase, but this argument needs to be made more clearly, and with clear comparison to a host of other works on robustness to paraphrases (or other flavors of rephrasing) including those on values (btw I could not find reference in the new text to the papers mentioned by the reviewers: ValueDCG and Do LLMs have Consistent Values).
The other issue is that it's not clear one really want absolute consistency for topics where there are nuances and arguments in both directions. Is it really a problem that an LLM would raise a certain argument when prompted in one way and another argument (possibly conflicting the stance of the previous argument) when prompted in another way. This is arguably something that a single human individual may also do. The authors are right that there is some level of contradiction that would make no sense, but if LLMs make such logical contradictions this isn't necessarily related to values per se. A related question, which the authors touch on, but should be clarified further, is to what degree we expect an LLM to act like a single person as opposed to a sample from a population.
In this context it would be interesting to do this evaluation when evaluating both pairs in the same context, so that one acts as the context for the other. In this context it might make more sense to expect consistency.

**Reviewer Concerns:**

The rebuttals addressed some of the reviewer concerns and one reviewer did acknowledge what as addressed and noted what's still missing.

**Reviewer Scores:**

I don't think the rating would have changed much. Maybe up by a point or so, but would still likely not support accepting.

---

### Decision · Program_Chairs · 2026-01-26

Reject